# RBD-specific antibody response after two doses of different SARS-CoV-2 vaccines during the mass vaccination campaign in Mongolia

**Burenjargal Batmunkh**[1‡], **Dashpagma Otgonbayar**[2,3‡], **Shatar Shaarii** [3⊙], **Nansalmaa Khaidav** [3⊙], **Oyu-Erdene Shagdarsuren**[3⊙], **Gantuya Boldbaatar**[4⊙], **Nandin-Erdene Danzan**[4⊙], **Myagmartseren Dashtseren**[4⊙], **Tsolmon Unurjargal**[4⊙], **Ichinnorov Dashtseren**[4⊙], **Munkhbaatar Dagvasumberel**[4⊙], **Davaalkham Jagdagsuren**[2⊙], **Oyunbileg Bayandorj**[2⊙], **Baasanjargal Biziya**[1⊙], **Seesregdorj Surenjid**[5⊙], **Khongorzul Togoo**[1⊙], **Ariunzaya Bat-Erdene**[1⊙], **Zolmunkh Narmandakh**[1⊙], **Gansukh Choijilsuren** [1⊙], **Ulziisaikhan Batmunkh**[1⊙], **Chimidtseren Soodoi**[1⊙], **Enkh-Amar Boldbaatar** [1⊙], **Ganbaatar Byambatsogt**[6⊙], **Otgonjargal Byambaa**[1⊙], **Zolzaya Deleg**[1⊙], **Gerelmaa Enebish**[1⊙], **Bazardari Chuluunbaatar**[7⊙], **Gereltsetseg Zulmunkh**[7⊙], **Bilegtsaikhan Tsolmon**[2⊙], **Batbaatar Gunchin**[1⊙], **Battogtokh Chimeddorj**[1⊙], **Davaalkham Dambadarjaa**[3⊙], **Tsogtsaikhan Sandag** [1]*

**1** School of Biomedicine, Mongolian National University of Medical Sciences, Sainshand, Mongolia, **2** National Center for Communicable Diseases of Mongolia, Ulaanbata, Mongolia, **3** School of Public Health, Mongolian National University of Medical Sciences, Sainshand, Mongolia, **4** School of Medicine, Mongolian National University of Medical Sciences, Sainshand, Mongolia, **5** International School of Mongolian Medicine, Mongolian National University of Medical Sciences, Sainshand, Mongolia, **6** School of Nursing, Mongolian National University of Medical Sciences, Sainshand, Mongolia, **7** Mongolia-Japan Hospital, Mongolian National University of Medical Sciences, Sainshand, Mongolia

⊙ These authors contributed equally to this work.
‡ BB and DO contributed equally to this work as Joint First Authors
* tsogtsaikhan.s@mnums.edu.mn

**Data Availability Statement:** There are ethical restrictions on publicly sharing the minimal data set for this study due to participant privacy

## Abstract

The SARS-CoV-2 vaccination campaign began in February 2021 and achieved a high rate of 62.7% of the total population fully vaccinated by August 16, 2021, in Mongolia. We aimed to assess the initial protective antibody production after two doses of a variety of types of SARS-CoV-2 vaccines in the Mongolian pre-vaccine antibody-naïve adult population. This prospective study was conducted from March-April to July-August of 2021. All participants received one of the four government-proposed COVID-19 vaccines including Pfizer/BioNTech (BNT162b2), AstraZeneca (ChAdOx1-S), Sinopharm (BBIBP-CorV), and Sputnik V (Gam-COVID-Vac). Before receiving the first shot, anti-SARS-CoV-2 S-RBD human IgG titers were measured in all participants (n = 1833), and titers were measured 21–28 days after the second shot in a subset of participants (n = 831). We found an overall average protective antibody response of 84.8% (705 of 831 vaccinated) in 21–28 days after two doses of the four types of COVID-19 vaccines. Seropositivity and titer of protective antibodies produced after two shots of vaccine were associated with the vaccine types, age, and residence of vaccinees. Seropositivity rate varied significantly between vaccine types, 80.0% (28 of 35) for AstraZeneca ChAdOx1-S; 97.0% (193 of 199) for Pfizer BNT162b2; 80.7% (474 of 587) for Sinopharm BBIBP-CorV, and 100.0% (10 of 10) for Sputnik V Gam-COVID-Vac, respectively. Immunocompromised vaccinees with increased risk for developing severe

concerns. Data are available upon request from the Corresponding Author, and from the Division of Science and Technology, Mongolian National University of Medical Sciences via email (sciencetechnology@mnums.edu.mn), or via phone (+976-7775-7575 (1010)), for researchers who meet the criteria for access to confidential data.

**Funding:** The authors received no specific funding for this work.

**Competing interests:** The authors have declared that no competing interests exist.

COVID-19 disease had received the Pfizer vaccine and demonstrated a high rate of seropositivity. A high geometric mean titer (GMT) was found in vaccinees who received BNT162b2, while vaccinees who received ChAdOx1-S, Sputnik V, and BBIBP-CorV showed a lower GMT. In summary, we observed first stages of the immunization campaign against COVID-19 in Mongolia have been completed successfully, with a high immunogenicity level achieved among the population with an increased risk for developing severe illness.

## Introduction

Mongolia had no local COVID-19 cases until November 2020, however from November 2020 to March 2021 local cases increased gradually [1,2]. The nationwide vaccination campaign started on 23 February 2021 in Mongolia. Mongolia received the first batch of the inactivated BBIBP-CorV vaccine from Sinopharm, China, and the non-replicating viral vector vaccine Oxford-AstraZeneca, from India and began immunization in high-risk healthcare workers [1,3]. Other priority groups, including the elderly and those with chronic illnesses, were vaccinated following the arrival of further doses. In addition, the mRNA vaccine, Pfizer-BioNTech, and the non-replicating viral vector vaccine Gamaleya's Sputnik V became available for administration later in the second half of 2021 in Mongolia [1]. The Mongolian campaign for a vaccination with these four types of vaccines was achieved at a high rate of 62.7% of the total population with full vaccination, as reported on August 16, 2021 [4,5].

Globally, the antibody production rate of these vaccines is well-documented [6–22]. However, we found few reports comparing the responses to multiple types of SARS-CoV-2 vaccines in the same vaccination campaign using the same antibody detection system. We explored the Mongolian experiences, with some unique experiences in this field including dispersed and rural populations. Despite successful vaccination campaigns, Mongolia has experienced an upsurge of new cases until February-March 2022, which may be related to the efficacy of the vaccine waning [5,23].

This study aimed to examine the initial protective antibody production after two doses of various types of SARS-CoV-2 vaccines in the Mongolian pre-vaccine antibody-naïve adult population.

## Materials and methods

### Study population

This prospective cohort study was conducted from April-May to July-August of 2021. All participants had received one of the four government-proposed COVID vaccines: Pfizer/BioNTech (BNT162b2), AstraZeneca (ChAdOx1-S), Sinopharm (BBIBP-CorV), and Sputnik V (Gam-COVID-Vac). Personal information and serum samples of vaccinees were collected in three population groups determined as a priority strategy by the Government of Mongolia.

The first group includes healthcare professionals and government employees (frontline employees) working in frontline places such as hospitals serving out and in-patients with confirmed SARS-CoV-2 infection, family doctors and emergency medical service units, and isolation campuses. Employees of randomly selected facilities in urban (a total of 14 sites including First State Hospital, Central Hospital of the Armed Forces, Mongolia-Japan Hospital of the Mongolian National University of Medical Sciences, and Family Medicine Centers) and rural healthcare sites (general hospitals and primary healthcare centers in 5 selected provinces-aimags) were presented in the group.

The second group includes people with increased risk for severe COVID-19, including immunocompromised patients after immunosuppressive therapy for cancer and systemic or autoimmune diseases (SAD), people living with human immunodeficiency virus infection and acquired immunodeficiency syndrome (PLWHA), pregnant women in the last two trimesters of pregnancy, and elderlies aged above 60 years. Cancer and SAD patients, and PLWHA were classified as the immunocompromised population. SAD patients were selected from patients observed at First State Hospital, cancer patients were selected from patients who are under observation at the National Cancer Center, and PLWHA were selected from patients observed at the National Center for Communicable Diseases. The elderly and pregnant women were selected from the vaccinees of the third group.

The third group represented the 18–59 years old, general adult population and was selected from vaccinees in 18 randomly selected vaccination units in Ulaanbaatar city. The second and third groups included attendants from Ulaanbaatar city only.

## Anti-SARS-CoV-2 RBD-IgG seroprevalence

We enrolled a total of 1864 vaccinees for measurement of SARS-CoV-2 receptor binding domain (RBD) immunoglobulin class G (IgG) and M (IgM) antibodies before the first dose of the COVID-19 vaccines as baseline. We then collected data and serum samples on days 21–28 after receiving the second dose of the vaccine from 831 vaccinees only (Fig 1).

A titer of anti-SARS-CoV-2 S-RBD human IgG (Proteintech®, USA) before the first dose and after the second dose administration of vaccines against COVID-19 was measured using Enzyme-linked Immunosorbent Assay (ELISA) in all participants. Anti-SARS-CoV-2 S-RBD human IgM (Proteintech®, USA) titer was measured at the same schedule and the measurement was used to determine ongoing or previous coronavirus infection. According to the manufacturer's instruction titer of anti-SARS-CoV-2 RBD-IgG $\geq$ 6.25 ng/mL and/or titer of anti-SARS-CoV-2 RBD-IgM antibody $\geq$ 6.25 ng/mL before the first dose administration were considered as previous or ongoing coronaviral infection [24,25]. We considered seroconversion when the ratio of IgG antibody titer measured after the second dose to those measured before vaccination was found equal to or higher than 4.0 and showed a titer value $\geq$ 6.25 ng/mL.

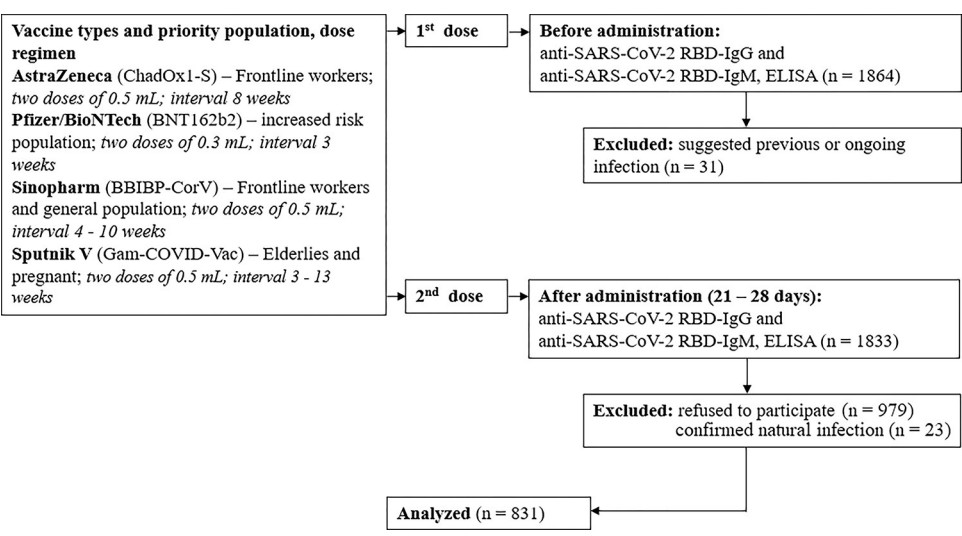

**Fig 1. Flowchart of postvaccine antibody response study.**

## Ethics statement

Study protocol and consent forms were reviewed and approved by the Ethical Review Committee under the Ministry of Health, under resolution no. 216, 217, and 219 from 6 April 2021.

## Statistical analysis

We performed both descriptive and inferential statistics. The distribution of seroprevalence among population subgroups was compared by Pearson's Chi-square test. The mean of variables, its standard deviation, and 95% confidence intervals were compared using analysis of variance (ANOVA). Protective antibody titer was compared using geometric mean titer (GMT) and geometric standard deviation (GSD). The predictive value of age for seroconversion was analyzed by receiver operating characteristics (ROC) analysis. In the ROC table, the Youden index (J) is calculated first, then the optimal cut-point (OCP) corresponds to the maximum value of the Youden index, sensitivity, and specificity of predictiveness. Logarithmic regression analysis was used for detecting an association between age and antibody titer. Statistical significance was expressed using p-values of $< 0.05$, $< 0.01$, $< 0.005$, and $< 0.001$.

# Results

## Study population

Sociodemographic, potential risk for developing severe illness, and vaccine-type information of study participants by two stages of observation are shown in the Table 1.

## Previous or ongoing infection

In 31 of 1864 vaccinees, we found previous (anti-SARS-CoV-2 RBD IgG $\geq$ 6.5 ng/mL; n = 15) or ongoing (anti-SARS-CoV-2 RBD IgG and anti-SARS-CoV-2 RBD IgM $\geq$ 6.5 ng/mL; n = 12) SARS-CoV-2 infection before the first dose and thus excluded from the analysis. The remaining 1883 vaccinees data were used for further analysis.

## Seroconversion rate after two doses of vaccine

We found at least a 4-fold increased titer of anti-SARS-CoV-2 RBD-IgG antibody in 705 (84.8%) of 831 vaccinees in 21–28 days (26.2 ± 3.3 days) after administration of the second dose of vaccine. We did not detect an increase of titer more than 4-fold in 97 vaccinees (15.2%) and classified them as non-responders. Seroconversion rates according to population groups and vaccine types are shown in Fig 2.

We did not find significant differences in seroconversion rates of population groups stratified by sex, age, permanent residence, and professional risk of medical and healthcare professionals and government employees working in the frontline (Fig 2A–2C and 2F). The seroconversion rate in urban and rural vaccinees was in the approximately same range, however, it was significantly varied according to aimags despite all rural residents being vaccinated with the same type of vaccine—Sinopharm BBIBP-CorV (Fig 2D). Seroconversion rates according to priority population for vaccination and risk for severe diseases varied significantly (Fig 2E and 2G). Sputnik V vaccine receivers have shown an absolute response of 100.0%, actually, we observed very few vaccinees in this group (n = 10). Pfizer BNT162b2 vaccine receivers have shown the high rate (97.0%) of positive response, while the population vaccinated with the Sinopharm BBIBP-CorV and the ChAdOx1-S vaccines demonstrated an approximately same a same rate of seroconversion (80.0% and 80.7%, respectively) (Fig 2H).

Vaccinees of different age groups did not show a significant variation in seroconversion rate (Fig 2B), however, ROC analysis of the age of vaccinees stratified by seroconversion

**Table 1. Sociodemographic, severe illness risk, and vaccine-type characteristics of study participants.**

| Characteristics of participants | Before 1st dose (n = 1864) | After 2nd dose (n = 831) |
|---|---|---|
| **Sex, count (percent)** | | |
| Males | 692 (37.3) | 314 (37.8) |
| Females | 1164 (62.7) | 517 (62.2) |
| **Age (years)** | | |
| Mean (M ± SD) | 40.9 ± 14.3 | 41.5 ± 14.0 |
| Median | 38.0 | 39.0 |
| CI 95 | 40.2–41.5 | 40.6–42.5 |
| Min.–Max. | 18–93 | 18–93 |
| **Age group, count (percent)** | | |
| < 20 | 31 (1.7) | 16 (1.9) |
| 20–29 | 403 (22.0) | 151 (18.2) |
| 30–39 | 554 (30.3) | 254 (30.6) |
| 40–49 | 376 (20.5) | 181 (21.8) |
| 50–59 | 261 (14.3) | 135 (16.2) |
| 60–69 | 130 (7.1) | 66 (7.9) |
| ≥ 70 | 75 (4.1) | 28 (3.4) |
| **Population groups, count (percent)** | | |
| **Urban*** | 1152 (63.0) | 520 (62.6) |
| **Rural (aimags)** | | |
| Bayankhongor | 119 (17.6) | |
| Bulgan | 128 (18.9) | 122 (39.2) |
| Darkhan-Uul | 70 (10.3) | 50 (16.1) |
| Dornod | 120 (17.7) | 97 (31.2) |
| Dundgovi | 121 (17.8) | |
| Orkhon | 120 (17.7) | 42 (13.5) |
| Subtotal† | 678 (37.0) | 311 (37.4) |
| **Frontline employees** | | |
| Employees working in "red-label" facilities‡ | 559 (60.0) | 150 (38.8) |
| Employees working in "yellow-label" facilities | 373 (40.0) | 237 (61.2) |
| Subtotal† | 932 (50.2) | 387 (46.6) |
| **Population with increased risk** | | |
| SAD^ | 134 (25.8) | 95 (42.0) |
| Cancer^ | 88 (17.0) | 20 (8.8) |
| PLWHA | 68 (13.1) | 60 (26.5) |
| Elderly | 119 (22.9) | 48 (21.2) |
| Pregnant | 110 (21.2) | 3 (1.3) |
| Subtotal† | 519 (28.0) | 226 (27.2) |
| **General population$†** | 405 (21.8) | 218 (26.2) |
| **Vaccine types** | | |
| AstraZeneca (ChAdOx1-S) | 147 (7.9) | 35 (4.2) |
| Pfizer/BioNTech (BNT162b2) | 403 (21.6) | 199 (23.9) |
| Sinopharm (BBIBP-CorV) | 1240 (66.5) | 587 (70.6) |

(*Continued*)

**Table 1.** (Continued)

| Characteristics of participants | Before 1st dose (n = 1864) | After 2nd dose (n = 831) |
|---|---|---|
| Sputnik V (Gam-COVID-Vac) | 74 (4.0) | 10 (1.2) |

*-residents of Ulaanbaatar city and suburban area

†-subgroup percentage was calculated within groups and the group subtotal percentage was calculated from the total number of participants

‡-employees working in direct contact with COVID-19 patients (included medical doctors, nurses, nurse assistants serving COVID-19 patients, radiologists, laboratory technicians collected samples, ambulance drivers and hospital porters, ward serving personnel, health officers from the emergency ward, and epidemiologists)

□-employees working without direct contact with patients (included police and security officers, officers of emergency service, personnel of hospital kitchen, inspectors, and administrative and service workers)

^-patients received corticosteroids due to systemic or autoimmune diseases and patients passed chemo- or radiation therapy due to solid cancer (time after last therapy < 6 months)

$-population aged 18–59 years; Abbreviations: M, mean; SD, standard deviation; CI95, confidence interval of 95%; Min.–Max., lowest and highest values; SAD, patients passed immunosuppressive therapy due to systemic or autoimmune diseases; PLWHA, People Living with Human Immunodeficiency Virus and Acquired Immunodeficiency Syndrome.

showed an increased probability of people aged $\geq$ 36 years to not respond to the vaccine exposure (Fig 3A). Furthermore, the age-dependent decline in seroconversion was significant for vaccinees who received the Sinopharm BBIBP vaccine (Fig 3B), but not those who received other types (p > 0.05).

## The titer of protective antibodies in vaccinees with the seroconversion

The mean titer of anti-SARS-CoV-2 RBD-IgG antibodies in vaccinees who showed seroconversion was 0.12 ± 0.15 ng/mL (CI95 0.11–0.13; median 0.1) before the vaccine administration and reached 111.8 ± 116.3 ng/mL (CI95 104.9–118.7; median 81.9) following to second dose. The geometric mean titer (GMT) of anti-SARS-CoV-2 RBD-IgG increased from 0.13 ± 0.09 ng/ml before vaccination to 69.9 ±17.6 ng/ml after two shots of vaccine, respectively. A comparison of GMT value after two doses of the vaccine in various population groups is shown in Table 2.

GMT demonstrated significant variation according to the sex, age, residence of vaccinees, targeted priority population, types of pathology or conditions increasing the risk for severe disease, and vaccine types. However, the sex and professional risk of frontline employees did not had a significant difference in GMT.

We studied the titer of protective antibodies in population groups of vaccinees who received BBIBP-CorV separately and found some significant associations. For instance, female vaccinees immunized with the Pfizer vaccine demonstrated higher levels of protective antibodies compared to males (Fig 4A), and the titer of protective antibodies was found associated with the age of vaccinees who received the Sinopharm vaccine (Fig 4B).

## Discussion

We found an average protective antibody response of 84.8% (in 705 of 831 vaccinees) in 21–28 days after two doses of the four types of COVID-19 vaccine. In our view, the following factors show an essential impact on postvaccine seroconversion. First, vaccine types likely played a crucial role in the seroprevalence. Seropositivity rate varied significantly by vaccine types showing 80.0% for AstraZeneca ChAdOx1-S; 97.0% for Pfizer BNT162b2; 80.7% for

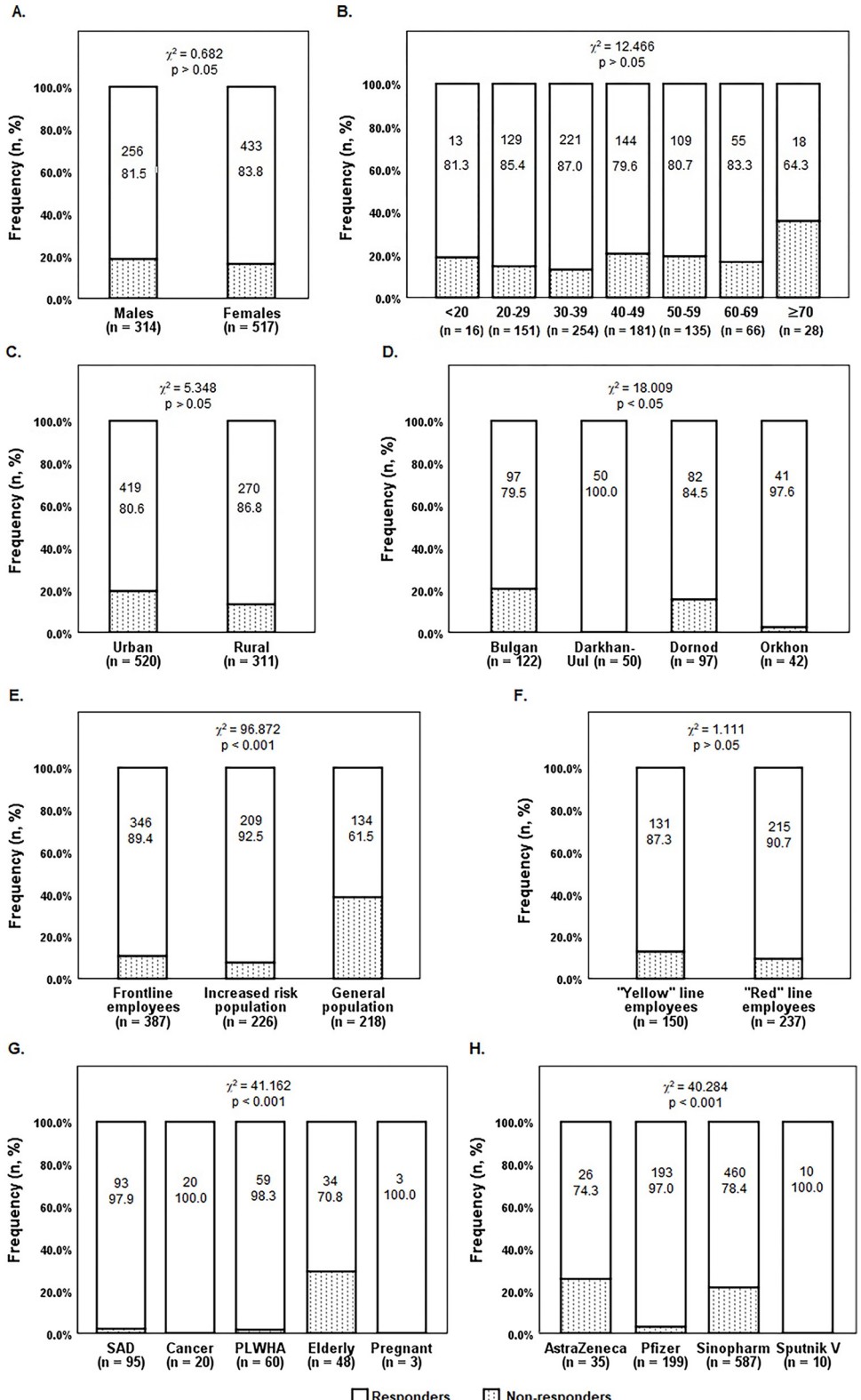

**Fig 2. Post-vaccine anti-SARS-CoV-2 RBD-IgG antibody response rate in various population groups.** A. Seroconversion rate according to the sex of vaccinees; B. Seroconversion rate according to the age of vaccinees; C and D. Seroconversion rate according to the residence of vaccinees; E. Seroconversion rate according to vaccination priority population (Here, Frontline workers–medical and health professionals and government employees working in

the frontline of a fight against infection; Increased risk population–immunocompromised and aged people who may develop severe COVID-19 disease in case of infection; and General population–people aged 18–59 years without increased risk); F. Seroconversion rate according to the professional risk of frontline employees; G. Seroconversion rate according to increased risk population subgroups; and H. Seroconversion rate according to vaccine types. Notes: n, count of vaccinees; %, percentage of vaccinees; p, asymptotic significance (two-sided); Abbreviation: SAD, patients received immunosuppressive therapy due to systemic or autoimmune disorders; PLWHA, people living with HIV and AIDS.

Sinopharm BBIBP-CorV, and 100.0% for Sputnik V Gam-COVID-Vac. Although immuno-compromised vaccinees from the population with increased risk for severe COVID-19 disease had received the Pfizer vaccine, 98.5% (192 out of 198) vaccinees in these cohorts demonstrated seropositivity. Seropositivity rates for SARS-CoV-2 (S) IgG after two doses of different types of vaccines are well-described, including AstraZeneca ChAdOx1-S (range 85.7–100.0%) [6,11,14,26,27], Pfizer BNT162b2 (range 93.6–100%) [19–22,28], Sinopharm BBIBP-CorV (range 60.6–99.2%) [6–12,14,15,17,21,27–29], and Sputnik V (range 94.5–100.0%) [27,28,30–32] vaccines.

Second, the age of vaccinees considerably affects seropositivity. We established 36 years as an optimal cut-point for seronegative state prediction. Among similarly designed studies, the majority of studies reported a lower seropositivity rate in elderly vaccines [13,20,22,26,33,34].

Third, the residence of vaccinees might play some role in seroconversion. For instance, in our study, rural residents demonstrated variable seropositivity and mean GMT according to aimags, although they received the same types of vaccines—Sinopharm (BBIBP-CorV).

A high GMT of anti-SARS-CoV-2 (S) IgG (192.5 ± 9.0 ng/mL) have found in vaccinees received Pfizer (BNT162b2) while vaccinees receiving AstraZeneca (ChAdOx1-S) and Sinopharm (BBIBP-CorV) demonstrated the lower GMT (88.2 ± 12.1, 63.4 ± 21 and 45.7 ± 15.6

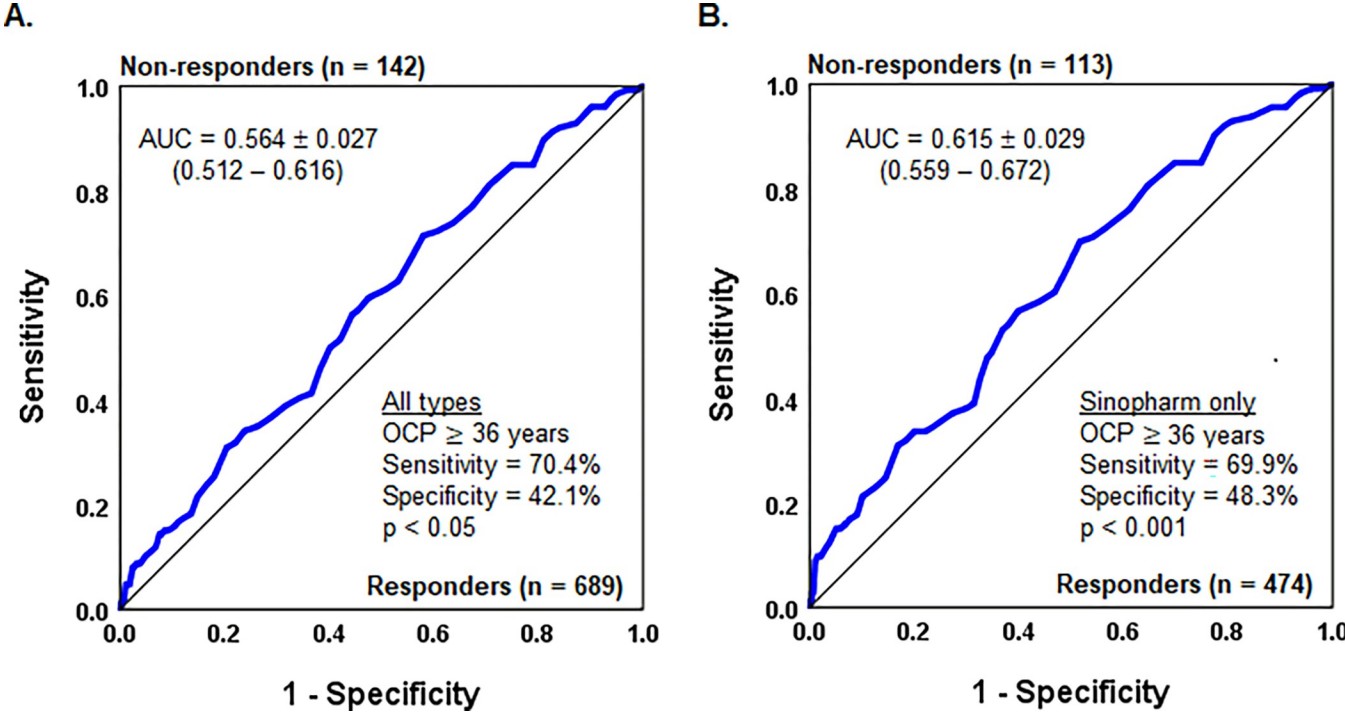

**Fig 3. ROC curve of the age of vaccinees stratified by the seroconversion.** A. ROC curve of the age of all vaccinees; B. ROC curve of age in vaccinees received the Sinopharm BBIBP vaccine. AUC, Area under the curve; OCP, optimal cut-point; p, asymptotic significance.

**Table 2. The geometric mean titer of anti-SARS-CoV-2 RBD-IgG antibody after two doses of vaccine against COVID-19 in vaccinees with seroconversion.**

| Population groups | | n | GMT ± GSD | Significance (p)* |
|---|---|---|---|---|
| **Total** | | 705 | 69.9 ± 17.6 | |
| **Sex** | Males | 264 | 65.4 ± 17.0 | < 0.05 |
| | Females | 441 | 72.8 ± 18.0 | |
| **Age groups** | < 20 | 15 | 43.6 ± 14.7 | < 0.05 |
| | 20–29 | 133 | 64.1 ± 16.5 | |
| | 30–39 | 222 | 68.9 ± 16.6 | |
| | 40–49 | 150 | 76.1 ± 17.3 | |
| | 50–59 | 111 | 74.9 ± 18.2 | |
| | 60–69 | 56 | 75.2 ± 22.4 | |
| | ≥ 70 | 18 | 60.5 ± 22.6 | |
| **Residence** | Urban | 435 | 85.6 ± 18.5 | < 0.001 |
| | Rural (aimags) | 270 | 50.5 ± 14.3 | |
| | Bulgan | 97 | 35.0 ± 12.4 | < 0.001 |
| | Darkhan-Uul | 50 | 68.4 ± 13.2 | |
| | Dornod | 82 | 49.7 ± 13.8 | |
| | Orkhon | 41 | 85.3 ± 13.7 | |
| **Priority groups** | Frontline workers | 346 | 58.2 ± 15.5 | < 0.001 |
| | Population with increased risk | 209 | 136.9 ± 16.5 | |
| | General adult population | 150 | 41.8 ± 14.7 | |
| **Professional risk of frontline employees** | Employees of "Red-label" facilities | 215 | 61.4 ± 14.3 | > 0.05 |
| | Employees of "Yellow-label" facilities | 131 | 56.4 ± 11.4 | |
| **Population with increased risk of developing severe disease** | SAD | 93 | 180.7 ± 11.5 | < 0.001 |
| | Cancer patients | 20 | 219.7 ± 7.9 | |
| | PLWHA | 59 | 165.6 ± 6.0 | |
| | Elderly | 34 | 34.5 ± 23.5 | |
| | Pregnant | 3 | 159.8 ± 8.0 | |
| **Vaccine types** | AstraZeneca (ChAdOx1-S) | 28 | 88.2 ± 12.1 | < 0.001 |
| | Pfizer (BNT162b2) | 193 | 192.5 ± 9.0 | |
| | Sinopharm (BBIBP-CorV) | 474 | 45.7 ± 15.6 | |
| | Sputnik V (Gam-COVID-Vac) | 10 | 63.4 ± 21.4 | |

*- statistical significance calculated using ANOVA. Abbreviations: Ng/mL, nanogram per millimeter; GMT, geometric mean titer; GSD, geometric standard deviation; SAD, patients passed immunosuppressive therapy due to systemic or autoimmune diseases; PLWHA, People Living with Human Immunodeficiency Virus and Acquired Immunodeficiency Syndrome.

ng/mL, respectively; p < 0.001). This finding was similar to the results of Sughayer MA (2022), who reported the highest anti-SARS-CoV-2 RBD-IgG response rate and mean titer in vaccinees received Pfizer BNT162b2 followed by vaccinees received AstraZeneca ChAdOx1-S and Sinopharm BBIBP-CorV [16].

We showed two population subgroups among vaccinees who received the BBIBP-CorV vaccine, namely vaccinees aged above 60 years and urban residents, may predict lower titer of postvaccination antibodies. Furthermore, in our study, we observed variable seropositivity and GMT among rural residents, depending on the aimags (provinces), despite receiving the same types of vaccines—Sinopharm (BBIBP-CorV). So far, we cannot give an exclusive explanation of this phenomenon since the sociodemographic pattern of these groups was approximately the same (S1 Table). Association of the seropositivity and titer of protective antibodies after two shot vaccination with sex, age, and residence of vaccinees were reported ambiguous

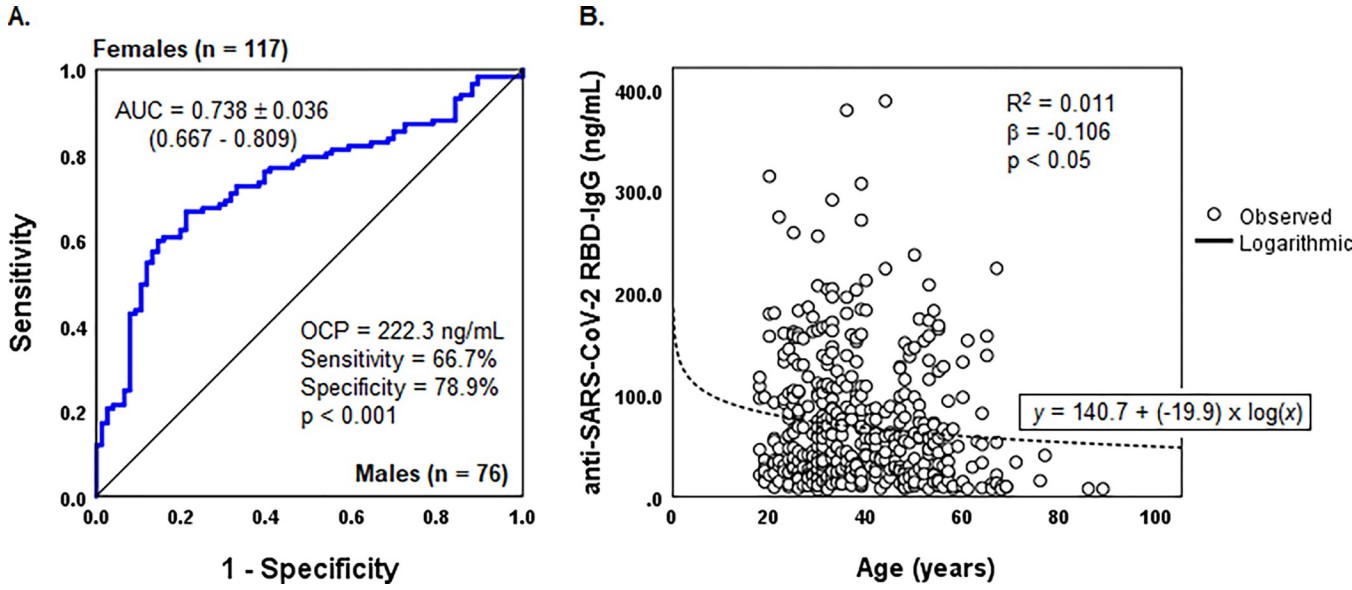

**Fig 4. Association of protective antibody titer with sex and age of vaccinees received certain types of vaccines.** A. ROC curves of anti-SARS-CoV-2 RBD-IgG titer stratified by the sex of vaccinees received the Pfizer vaccine; B. Logarithmic regression between anti-SARS-CoV-2 RBD-IgG titer and age of vaccinees received Sinopharm vaccine. Abbreviations: AUC, Area Under the Curve; OCP, optimal cut-point; p, asymptotic significance; $R^2$, determination coefficient; β, standardized coefficient of regression; p, statistical significance (ANOVA).

[35,36]. Age was often reported as a predictor of seropositivity in subjects who vaccinated with the BBIBP-CorV vaccine [13,35,37]. We found a 98.2% seropositivity after two shots of the BNT162b vaccine among immunocompromised individuals, including patients who received immunosuppressive therapy and PLWHA. In contrast, seropositivity rates of 77.0–85.2% were reported in cohorts of immunocompromised patients in UK [38], Turkey [34], and USA [39].

## Conclusion

In summary, we demonstrate a successful accomplishment of the first stage of the immunization campaign against COVID-19 in Mongolia, with certain high immunogenicity levels found in the population with increased risk for severe disease. Seropositivity and titer of protective antibodies produced following two shots of the SARS-CoV-2 vaccine were associated with the vaccine types, age, and residence of vaccinees.

## Study limitation and considerations for further study

Data and serum collection time after two doses of vaccine in this study timely overlayed with the period of strict lockdown measures in the country. Many participants refused further participation in the study because of fear of being infected. For this reason, data and serum samples after complete vaccination were available only yielded in 831 (44.6%) of 1864 eligible vaccinees. However, we suggest our baseline data will be pivotal for a further study concerning the morbidity of vaccinees registered later.

## Supporting information

**S1 Table. The sociodemographic pattern of frontline employees from different rural sites.** (DOCX)

## Acknowledgments

We thank the Ministry of Health of Mongolia for its support in conducting the study.

## Author Contributions

**Conceptualization:** Gantuya Boldbaatar, Davaalkham Dambadarjaa.

**Data curation:** Burenjargal Batmunkh, Dashpagma Otgonbayar, Shatar Shaarii, Nansalmaa Khaidav, Oyu-Erdene Shagdarsuren, Nandin-Erdene Danzan, Myagmartseren Dashtseren, Davaalkham Jagdagsuren, Oyunbileg Bayandorj, Khongorzul Togoo, Ganbaatar Byambat-sogt, Otgonjargal Byambaa, Zolzaya Deleg, Gerelmaa Enebish, Bazardari Chuluunbaatar, Tsogtsaikhan Sandag.

**Formal analysis:** Burenjargal Batmunkh, Dashpagma Otgonbayar, Tsogtsaikhan Sandag.

**Investigation:** Burenjargal Batmunkh, Dashpagma Otgonbayar, Shatar Shaarii, Oyu-Erdene Shagdarsuren, Gantuya Boldbaatar, Nandin-Erdene Danzan, Myagmartseren Dashtseren, Davaalkham Jagdagsuren, Khongorzul Togoo, Ariunzaya Bat-Erdene, Zolmunkh Narman-dakh, Gansukh Choijilsuren, Ulziisaikhan Batmunkh, Chimidtseren Soodoi, Enkh-Amar Boldbaatar, Gereltsetseg Zulmunkh, Tsogtsaikhan Sandag.

**Methodology:** Burenjargal Batmunkh, Dashpagma Otgonbayar, Shatar Shaarii, Tsolmon Unurjargal, Ichinnorov Dashtseren, Davaalkham Jagdagsuren, Baasanjargal Biziya, Seesreg-dorj Surenjid, Khongorzul Togoo, Ariunzaya Bat-Erdene, Zolmunkh Narmandakh, Gan-baatar Byambatsogt, Bilegtsaikhan Tsolmon, Batbaatar Gunchin, Battogtokh Chimeddorj, Davaalkham Dambadarjaa, Tsogtsaikhan Sandag.

**Project administration:** Dashpagma Otgonbayar, Munkhbaatar Dagvasumberel, Batbaatar Gunchin, Battogtokh Chimeddorj, Davaalkham Dambadarjaa, Tsogtsaikhan Sandag.

**Resources:** Munkhbaatar Dagvasumberel.

**Supervision:** Bilegtsaikhan Tsolmon, Batbaatar Gunchin, Battogtokh Chimeddorj, Davaalk-ham Dambadarjaa, Tsogtsaikhan Sandag.

**Validation:** Tsogtsaikhan Sandag.

**Visualization:** Tsogtsaikhan Sandag.

**Writing – original draft:** Tsogtsaikhan Sandag.

**Writing – review & editing:** Tsogtsaikhan Sandag.

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
