## [Decision Letter · Decision Letter 0]

23 Jun 2023

PONE-D-23-03663RBD-specific antibody response after two doses of different SARS-CoV-2 vaccines during the mass vaccination campaign in MongoliaPLOS ONE

Dear Dr. Tsogtsaikhan Sandag,

Thank you for submitting your manuscript to PLOS ONE. After careful consideration, we feel that it has merit but does not fully meet PLOS ONE’s publication criteria as it currently stands. Therefore, we invite you to submit a revised version of the manuscript that addresses the points raised during the review process.

We look forward to receiving your revised manuscript.

Kind regards,

Paavani Atluri

Academic Editor

PLOS ONE

Journal Requirements:

a) Did participants provide their written or verbal informed consent to participate in this study?

 Whilst you may use any professional scientific editing service of your choice, PLOS has partnered with both American Journal Experts (AJE) and Editage to provide discounted services to PLOS authors. Both organizations have experience helping authors meet PLOS guidelines and can provide language editing, translation, manuscript formatting, and figure formatting to ensure your manuscript meets our submission guidelines. To take advantage of our partnership with AJE, visit the AJE website (http://aje.com/go/plos) for a 15% discount off AJE services. To take advantage of our partnership with Editage, visit the Editage website (www.editage.com) and enter referral code PLOSEDIT for a 15% discount off Editage services. If the PLOS editorial team finds any language issues in text that either AJE or Editage has edited, the service provider will re-edit the text for free.

 A clean copy of the edited manuscript (uploaded as the new *manuscript* file).

Reviewers' comments:

Reviewer's Responses to Questions

**Comments to the Author**

1. Is the manuscript technically sound, and do the data support the conclusions?

Reviewer #1: Yes

Reviewer #2: Partly

2. Has the statistical analysis been performed appropriately and rigorously? 

Reviewer #1: Yes

Reviewer #2: Yes

3. Have the authors made all data underlying the findings in their manuscript fully available?

Reviewer #1: Yes

Reviewer #2: Yes

4. Is the manuscript presented in an intelligible fashion and written in standard English?

Reviewer #1: Yes

Reviewer #2: Yes

5. Review Comments to the Author

Reviewer #1: I find all the parts of the paper correctly written from background to the conclusion and of high quality. Methods are clearly explained. Sample size is proper for this kind of research and enables conclusions to the wider population. Results are clear and correct. Statistical analyses is proper for this kind of research. Discussion is sound and comments various research with comparison to findings of this paper. Limitations are clearly stated. All the data needed to understand the paper are presented in clear and common fashion. English is standard and paper is presented well. I can only extend my congratulations to the authors and my gratitude to the editor for the possibility to review this paper. I have no suggestions to improve this paper.

Reviewer #2: Thank you for inviting me to review this very interesting manuscript, which provides insight into the vaccination roll out in Mongolia. This study shows more granular clinical information than prior studies in this country and focuses specifically on those who are 'naive' to prior infection. Given these two novel features there are some major considerations:

1. Which type of vaccine an individual received was based on which group they were in. High-risk healthcare workers received BBIBP-CorV. Priority groups received vaccination ‘following the arrival of further doses’, presumably BBIBP? (Note in discussion it says these some of this group received Pfizer - Line 189). Pfizer, ChAdOx1, Sputnik became available later when vaccination rollout was expanded to the whole population. Therefore comparison of types of vaccine has major confounders including age (high risk included those over 65 years old) and co-morbidities. These confounders are well described as impacting antibody response, yet the analysis does not account for this when comparing antibody response by vaccine type. It is significantly understated in the authors discussion, which is surprising.

2. The study cohort have been selected on having no prior infection to the first dose. It is unclear how the authors accounted for infection between vaccinations, which again will influence antibody response. If the authors have N-antibody available, they should include this in their analysis. If they don't, this should again be discussed in detail in their discussion and consider how this may impact their findings (e.g., probability of infection in that time window based on transmission risk at the time - note some of this study was conducted during a lockdown).

3. It would be useful to know how exactly participants in the third group were selected (e.g., household based, clinic based...etc). How they were selected, and how it differs to the other groups, may highlight selection bias and would need to be explored in the discussion. I presume selection of group 2 was base done on hospital records? Again, this is not clear.

More specific queries include:

Line 48: Unclear what ‘locally grown cases’ means? Cases that had no clear link to migration?

Table 1: What are ‘red-line’ and ‘yellow-line’ facilities? Can you please provide explanation in caption?

Table 2: Please include use of ANOVA in the statistical analysis part of methods.

Line 210: Can you clarify definition of ‘sociodemographic’ pattern please? It would be interesting to know age distribution, proportion of sex and proportion of co-morbidities between the rural areas and rural vs urban.

I hope my comments has been useful and apologies in advance if I overlooked answers to my queries that are already in the manuscript.

6. PLOS authors have the option to publish the peer review history of their article (what does this mean?). If published, this will include your full peer review and any attached files.

Reviewer #1: **Yes: **Vladimir Petrovic

Reviewer #2: **Yes: **Annalan Mathew Dwight Navaratnam

---

## [Author Response · Author response to Decision Letter 0]

13 Jul 2023

Firstly, I would like to inform you that we analyzed all data thoroughly after we submitted the manuscript in early February 2023. And we found 311 participants who were excluded from the analysis due to “not complete” data in primary research units. Many participants were excluded due to insufficient personal information (e.g., unclear ID number, home address, and change in health condition after the first dose). Once we had enough time and no covid restrictions, we connected with participants and clarified their missing personal information. Finally, we added data from 254 participants for the analysis. As a result, some values of our study have to be changed. But these changes were not principal, and the main conclusions remained unchanged. 

Answer to Review Comments 

Reviewer #1: I find all the parts of the paper correctly written from background to the conclusion and of high quality. Methods are clearly explained. Sample size is proper for this kind of research and enables conclusions to the wider population. Results are clear and correct. Statistical analyses is proper for this kind of research. Discussion is sound and comments various research with comparison to findings of this paper. Limitations are clearly stated. All the data needed to understand the paper are presented in clear and common fashion. English is standard and paper is presented well. I can only extend my congratulations to the authors and my gratitude to the editor for the possibility to review this paper. I have no suggestions to improve this paper.

Answer: Thank you

Reviewer #2: Thank you for inviting me to review this very interesting manuscript, which provides insight into the vaccination roll out in Mongolia. This study shows more granular clinical information than prior studies in this country and focuses specifically on those who are 'naive' to prior infection. Given these two novel features there are some major considerations:

1. Which type of vaccine an individual received was based on which group they were in. High-risk healthcare workers received BBIBP-CorV. Priority groups received vaccination ‘following the arrival of further doses’, presumably BBIBP? (Note in discussion it says these some of this group received Pfizer - Line 189). Pfizer, ChAdOx1, Sputnik became available later when vaccination rollout was expanded to the whole population. Therefore comparison of types of vaccine has major confounders including age (high risk included those over 65 years old) and co-morbidities. These confounders are well described as impacting antibody response, yet the analysis does not account for this when comparing antibody response by vaccine type. It is significantly understated in the authors discussion, which is surprising.

Answer:

Date of arrivals for four types of vaccines:

AstraZeneca - February 23, 2021, and March 12, 2023.

https://www.who.int/mongolia/news/detail/23-02-2021-covid-19-vaccination-rollout-in-mongolia

https://www.unicef.org/mongolia/press-releases/mongolia-welcomes-first-batch-covid-19-vaccines-covax-facility

Sinopharm - February 23, 2021, and April 20, 2021

https://thediplomat.com/2021/05/how-mongolia-made-the-most-of-vaccine-diplomacy

https://asia.nikkei.com/Spotlight/Coronavirus/COVID-vaccines/Mongolia-resumes-vaccinations-as-worst-outbreak-rolls-on

Sputnik V - February 27, 2021, and April 30, 2021

https://news.mn/en/795436/

https://www.capitalsinitiative.org/2021/06/15/covid-19-vaccine-rollout-accelerated-to-all-above-18-years-eligible-in-ulaanbaatar/

Pfizer - June 17, 2021

https://www.unicef.org/mongolia/press-releases/over-84000-pfizer-vaccines-arrive-mongolia-first-batch-25-million-doses#:~:text=Press%20release-,Over%2084%2C000%20Pfizer%20vaccines%20arrive%20in%20Mongolia%2C%20the,batch%20of%202.5%20million%20doses.&text=June%2016%2C%202021%20Ulaanbaatar%20%2D%20Over,batch%20of%202.5%20million%20doses. 

As you see, AstraZeneca and Sinopharm vaccines arrived before started mass vaccination, on February 23, 2021. Frontline workers started the vaccination with AstraZeneca, but there were only 14.4 thousand doses of this vaccine. The arrival of the next batch of AstraZeneca was delayed. The Ministry of Health decided to continue the vaccination of frontline employees with the Sinopharm vaccine. So, we have participants who received both vaccines in this group. It is my fault; I did not check the citation source accurately. I corrected the text as follows: “Mongolia received the first batch of the inactivated BBIBP-CorV vaccine from Sinopharm, China, and the non-replicating viral vector vaccines Oxford-AstraZeneca, from India and began immunization in high-risk healthcare workers” 

Choice of vaccine type

Citizens of Mongolia had the opportunity to choose the type of vaccine that was available in the immunization unit.

Age and comorbidity impact 

All immunocompromised patients or patients who passed immunosuppressive therapy (SAD and Cancer patients) and PLWHA were immunized only with the Pfizer vaccine. 32 out of 48 elderlies received the Sinopharm vaccine. 

Additionally, I have performed ROC analysis of age by seroconversion in vaccine-type groups and found a significant impact only in the Sinopharm group. This finding was added to Figure 3 and explained in the text.

2. The study cohort have been selected on having no prior infection to the first dose. It is unclear how the authors accounted for infection between vaccinations, which again will influence antibody response. If the authors have N-antibody available, they should include this in their analysis. If they don't, this should again be discussed in detail in their discussion and consider how this may impact their findings (e.g., probability of infection in that time window based on transmission risk at the time - note some of this study was conducted during a lockdown).

Answer:

I agree with you. We did not provide serological or PCR evidence for infection between shots. The serological investigation was not rational because the Sinopharm vaccine is able to stimulate the production of N-antibodies. We used the following two database to exclude cases of natural infection: 1) we collected information regarding possible natural infections which may occur during the observation by filling out a questionnaire when vaccinees were invited for data and sample collection in days 21 – 28 after second dose. 2) In Mongolia, all new PCR confirmed cases of SARS-CoV-2 infection were registered in the “Gerege” system https://gerege.mn/en/home. This system was connected directly to the database of the Ministry of Health. We have checked all suspicious cases in this database later for exclusion of natural infection from the analysis. 

I have added a correction in Figure 1 regarding your comment.

3. It would be useful to know how exactly participants in the third group were selected (e.g., household based, clinic based...etc). How they were selected, and how it differs to the other groups, may highlight selection bias and would need to be explored in the discussion. I presume selection of group 2 was base done on hospital records? Again, this is not clear.

Answer:

Participants in the third group were randomly selected in family doctor’s centers. Vaccine administration units were organized in family doctor centers. Family doctor’s units were randomly selected in two of six urban districts of Ulaanbaatar city. Ulaanbaatar is divided into nine düüregs (usually translated as districts, six districts located in the urban areas, but three districts located in suburban areas), which are further subdivided into khoroos (most often translated as subdistricts). link: https://en.wikipedia.org/wiki/Administrative_divisions_of_Mongolia#:~:text=Ulaanbaatar%20is%20divided%20into%20nine,%2C%20microdistrict%20or%20simply%20district). An explanation according to a selection of participants from group two and three was done in the Methods section.

More specific queries include:

Line 48: Unclear what ‘locally grown cases’ means? Cases that had no clear link to migration?

Answer:

This phrase was cited directly from the reference. I corrected it as “local cases increased gradually”.

Table 1: What are ‘red-line’ and ‘yellow-line’ facilities? Can you please provide explanation in caption?

Answer:

I replaced the word “line” with the word “label”. An explanation of the employees of “red label” and “yellow label” facilities was updated in the caption of the Table 1.

Table 2: Please include use of ANOVA in the statistical analysis part of methods.

Answer: 

The use of ANOVA was included in the Statistical Analysis subsection of the Methods section.

Line 210: Can you clarify definition of ‘sociodemographic’ pattern please? It would be interesting to know age distribution, proportion of sex and proportion of co-morbidities between the rural areas and rural vs urban.

Answer: 

Here, the term “sociodemographic pattern” was used for frontline employees of different rural sites. This data was shown in newly added S1 Table regarding your comment.

---

## [Decision Letter · Decision Letter 1]

10 Nov 2023

PONE-D-23-03663R1RBD-specific antibody response after two doses of different SARS-CoV-2 vaccines during the mass vaccination campaign in MongoliaPLOS ONE

Dear Dr. Sandag,

Thank you for submitting your manuscript to PLOS ONE. After careful consideration, we feel that it has merit but does not fully meet PLOS ONE’s publication criteria as it currently stands. Therefore, we invite you to submit a revised version of the manuscript that addresses the points raised during the review process.

We look forward to receiving your revised manuscript.

Kind regards,

Ashraful Hoque

Academic Editor

PLOS ONE

Reviewers' comments:

Reviewer's Responses to Questions

**Comments to the Author**

1. If the authors have adequately addressed your comments raised in a previous round of review and you feel that this manuscript is now acceptable for publication, you may indicate that here to bypass the “Comments to the Author” section, enter your conflict of interest statement in the “Confidential to Editor” section, and submit your "Accept" recommendation.

Reviewer #1: All comments have been addressed

Reviewer #3: (No Response)

2. Is the manuscript technically sound, and do the data support the conclusions?

Reviewer #1: Yes

Reviewer #3: Yes

3. Has the statistical analysis been performed appropriately and rigorously? 

Reviewer #1: Yes

Reviewer #3: Yes

4. Have the authors made all data underlying the findings in their manuscript fully available?

Reviewer #1: (No Response)

Reviewer #3: Yes

5. Is the manuscript presented in an intelligible fashion and written in standard English?

Reviewer #1: Yes

Reviewer #3: Yes

6. Review Comments to the Author

Reviewer #1: I had no comments nor asked for revision. Initial version of the manuscript was already enough to recommend this paper to be accepted. So, i stay with my previous text on the same question.

I find all the parts of the paper correctly written from background to the conclusion and of high quality. Methods are clearly explained. Sample size is proper for this kind of research and enables conclusions to the wider population. Results are clear and correct. Statistical analyses is proper for this kind of research. Discussion is sound and comments various research with comparison to findings of this paper. Limitations are clearly stated. All the data needed to understand the paper are presented in clear and common fashion. English is standard and paper is presented well. I can only extend my congratulations to the authors and my gratitude to the editor for the possibility to review this paper. I have no suggestions to improve this paper.

Answer: Thank you

Reviewer #3: Manuscript shows interesting data related to the COVID-19 immunization in Mongolia, including organization of the vaccination campaign, coverage, seroconversion rates and antibody levels after different types of vaccines and in different groups.

All parts of the manuscript are written in intelligible fashion. Manuscript organization, structure, style and format are in accordance with the submission guidelines.

Introduction provides clear background. Methods, sample size, statistical analyses - all are appropriate and clearly explained. Results are clear. Discussion includes all sound and comments various research with comparison to findings of this paper. Limitations are clearly defined.

In the discussion part, it would be interesting to hear the author's thoughts on the possible reasons for the different seroconversion rates and mean GMT between different aimags, as well as lower titer in urban residents.

The authors provided detailed answers to the reviewers and added all suggested corrections into the appropriate parts of the manuscript.

7. PLOS authors have the option to publish the peer review history of their article (what does this mean?). If published, this will include your full peer review and any attached files.

Reviewer #1: No

Reviewer #3: No

---

## [Author Response · Author response to Decision Letter 1]

14 Nov 2023

Response to Reviewers was made in Revision 3. No special comments were done for last revision.

---

## [Editor Report · Decision Letter 2]

17 Nov 2023

RBD-specific antibody response after two doses of different SARS-CoV-2 vaccines during the mass vaccination campaign in Mongolia

PONE-D-23-03663R2

Dear Dr. Sandag,

We’re pleased to inform you that your manuscript has been judged scientifically suitable for publication and will be formally accepted for publication once it meets all outstanding technical requirements.

Kind regards,

Lyra Lynn Cauman

Support Staff - Editorial

PLOS ONE
---

## [Editor Report · Acceptance letter]

30 Nov 2023

PONE-D-23-03663R2 

RBD-specific antibody response after two doses of different SARS-CoV-2 vaccines during the mass vaccination campaign in Mongolia 

Dear Dr. Sandag:

I'm pleased to inform you that your manuscript has been deemed suitable for publication in PLOS ONE. Congratulations! Your manuscript is now with our production department. 

Kind regards, 

on behalf of

Dr. Ashraful Hoque 

Academic Editor

PLOS ONE